

# Population and distribution of wild Asian elephants (*Elephas maximus*) in Phu Khieo Wildlife Sanctuary, Thailand

Nyi Nyi Phyo Htet[1], Rattanawat Chaiyarat[2], Nikorn Thongthip[3], Panat Anuracpreeda[4], Namphung Youngpoy[2] and Phonlugsamee Chompoopong[5]

[1] Faculty of Environment and Resource Studies, Mahidol University, Salaya, Nakhon Pathom Province, Thailand
[2] Wildlife and Plant Research Center, Faculty of Environment and Resource Studies, Mahidol University, Salaya, Nakhon Pathom Province, Thailand
[3] Faculty of Veterinary Medicine, Kasetsart University, Kampang Saen, Nakhon Pathom Province, Thailand
[4] Institute of Molecular Biosciences, Mahidol University, Salaya, Nakhon Pathom Province, Thailand
[5] Phu Khieo Wildlife Sanctuary, Department of National Parks, Wildlife and Plant Conservation, Khon San District, Chaiyaphum Province, Thailand

## ABSTRACT

**Background**. The populations of wild Asian elephants (*Elephas maximus*) have increased recently after a period of worldwide decline in protected areas. It is important to understand the dynamics and distribution of the remaining populations to ensure their conservation and prevent human-elephant conflicts.

**Methods**. We monitored the population distribution of elephants between 2016 and 2019 in the Phu Khieo Wildlife Sanctuary, Thailand. We set one hundred forty-nine camera trap locations; cameras recorded 38,834 photos over 6,896 trap nights. Elephants were captured in 4,319 photographs. The maximum entropy modeling software MaxEntwas used to identify elephants' habitat preferences within 49 of the 149 total camera trap locations according to five environmental factors.

**Results**. One hundred fourteen elephants were identified. We identified 30 adult males, 43 adult females, 14 sub-adult males, nine sub-adult females, 11 juveniles, and seven calves. The age structure ratio based on adult femaleswas 0.7:1:0.3:0.2:0.3:0.2, and the ratio of reproductive ability between adult females, juveniles, and calves was 1:0.2:0.1. A suitable elephant habitat was determined to be 1,288.9 km$^2$ using Area Under the Curve (AUC). An AUC = 0.061 indicated good performance. Our model classified habitat preferences associated with elevation, forests, salt licks, human activity, and slope.

**Conclusions**. According to our probability map this sanctuary can provide a suitable habitat for elephants. Our results indicate that effective management practices can protect wild Asian elephants in the region and reduce conflict between humans and elephants.

Corresponding author
Rattanawat Chaiyarat,
rattanawat.cha@mahidol.ac.th

## INTRODUCTION

Wild Asian elephants (*Elephas maximus*) are the largest living terrestrial mammals in Asia. Wild elephants play a crucial role in forest ecosystem by opening up forests and distributing the seeds of trees and shrubs from one place to another. Because of this, they are commonly referred to as an umbrella species (*Tan et al., 2021*). Asian elephants are found in grasslands, tropical evergreen forests, semi-evergreen forests, moist deciduous forests, and dry deciduous forests in 13 countries (*Choudhury, 1999*). In the past, habitat loss was a primary factor in the decline of the species. As the human population in the region has increased, vast areas of this elephant's forest habitat were logged or converted for agriculture. This isolated elephants in habitat patches as ancient migratory routes were cut off (*Acharya et al., 2016*). Wild Asian elephant populations are also threatened by ivory and game hunters (*Vigne, 2013*; *Prakash et al., 2020*). Consequently, between 2003 and 2020, this elephant population declined from an estimated 41,410–52,345 individuals (*Sukumar, 2003*) to approximately 4,189–6,999 individuals (*Williams et al., 2020*). According to the International Union for Conservation of Nature (IUCN) red list, the wild elephant is endangered in each country worldwide (*IUCN, 2020*).

In Thailand, wild Asian elephants are spread across protected areas, mainly in the mountains along the border with Myanmar. Elephants are also found in smaller fragmented populations in the southern peninsula; several forest complexes on the border with Malaysia; to the east in a forest complex made up of the Khao Ang Runai Wildlife Sanctuary, Khao Soi Dao Wildlife Sanctuary, Khao Khitchakut National Park, and Khao Cha Mao National Park; and to the northeast at the Dong Phaya Yen-Khao Yai Forest Complex, which includes Khao Yai National Park, and the Western Isaan Complex. The degradation and fragmentation of elephant habitats are the biggest threat to Thailand (*Suksavate, Duengkae & Chaiyes, 2019*) as these increase conflicts between humans and elephants (*vandeWater & Matteson, 2018*). The population of wild elephants in Thailand in 2020 is made up of only approximately 3,126 to 3,341 individuals (*Williams et al., 2020*).

Elephants' habitats are fragmented in the protected areas, but they also include agricultural areas, which increase human-elephant conflicts near the sanctuary (*Chaiyarat, Youngpoy & Prempree, 2015*). It is important to understand the distribution and dynamics of the remaining populations to ensure effective conservation practices and prevent conflict.

## MATERIALS & METHODS

### Study area

Our study was conducted from 2016 to 2019 in the Phu Khieo Wildlife Sanctuary (PKWS) which spans over 1,560 km$^2$ in the Chaiyaphum Province of Thailand (latitude 16°5′ and 16°35′N and longitude 101°20′ and 101°55′E) (Fig. 1). Eight connecting protected areas cover more than 4,594 km$^2$ in the Western Issan Forest Complex. The sanctuary is near three other protected areas: the Nam Nao National Park to the north, Tat Mok National Park to the west, and Ta Bao-Huai Yai Wildlife Sanctuary to the southwest. The annual rain fall is 1,368 mm per year, and the average temperature is approximately 18 °C to

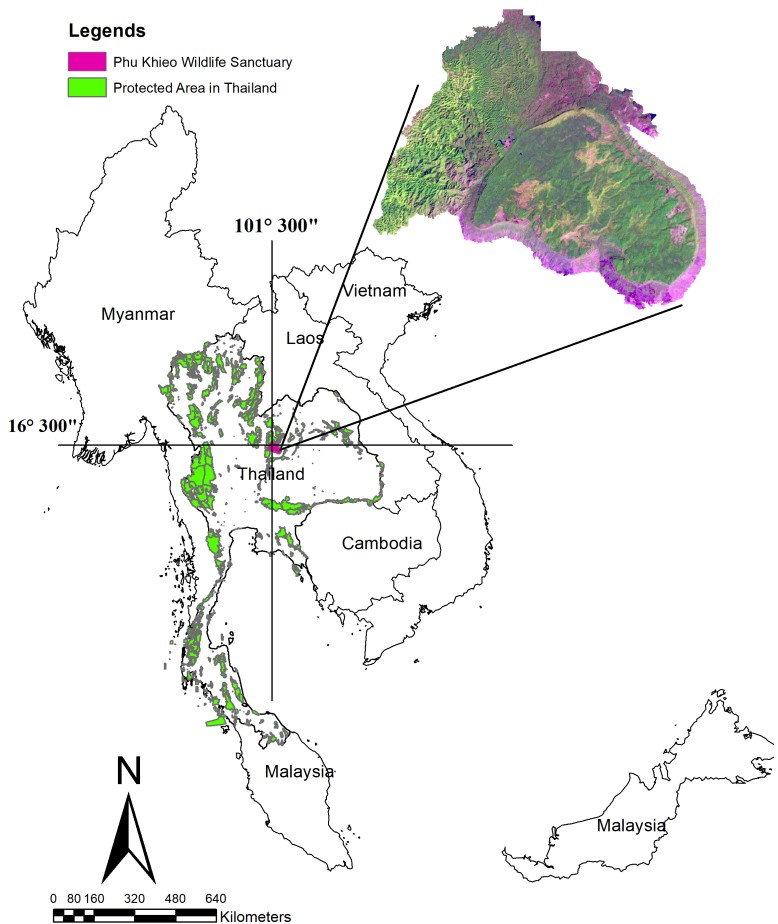

**Figure 1** Location of the Phu Khieo Wildlife Sanctuary, Thailand.

27 °C. The average elevation is 900 m (min. 200 m to max. 1,300 m) above the average sea level (ASL). Dry evergreen forests cover approximately 68% of the area. Of this total area, 27% are mixed deciduous forests dominating the low lands, 4% are dry dipterocarp forests, 0.6% are pine forests, and 0.4% area secondary forest (*Faculty of Forestry, 2010*).

## Camera trap survey

Global Positioning System (GPS) coordinate information was collected from camera trap placements in the field. The environmental conditions were the same ones recorded by *Pearce & Boyce (2006)*. A set of environmental factors that likely model a species' environmental requirements was obtained from a set of occurrence localities, influencing the suitability of the environment for the species (*Phillips, 2017*). Seven environmental factors were combined with the five predicted factors to generate the model. These were: human activity (road, stream, wildlife guard station, and villages), forest types, salt licks, elevation, and slope. As previously reported in *Chaiyarat et al. (2019)*, environmental parameters generated from available Geographic Information System (GIS) layers and habitat composition was analyzed using the land-use layers of the Phu Khieo Wildlife

Sanctuary digital map. Topography data was obtained from a digital elevation model (DEM) generated by the *Faculty of Forestry (2010)* from 1:50,000 topographic maps. A DEM was used to generate the slope and then images were resampled to a 30-m pixel resolution (Fig. 2). These parameters were used to estimate the distance between parameters in each pixel and each elephant observation point. A model built using 49 camera sites captured wild Asian elephants in 4,319 photographs. The other 100 camera trap stations were considered pseudo-absent from the total 149 camera trap stations, which captured 38,834 photographs. Images were captured using MaxEnt set to a 25 random test percentage with 15 replicates (more than the sample size) and 5,000 maximum iterations.

We used the HCO SG565 flash camera-traps (*HCO Outdoor Products, Norcross, Georgia, USA*) to obtain photographs of individual elephants. Fifteen camera trap stations systematically were set up in $3 \times 3$ km$^2$ grid cells within 3 months of sampling blocks. Two camera traps were used per grid with 10 m apart from each other. The camera traps operated continually for 24 h a day, recording the date and time of each photograph. The batteries were changed in the camera traps monthly, and the SD card data was transferred to HD data storage. Camera traps were attached to trees approximately 0.75 m from the ground (*Rowcliffe et al., 2008*) with a view range of at least five m to six meters to capture a single wild elephant up to 20 m away to allow for a complete view of a wild elephant herd (*Varma, Pittet & Jamadagni, 2006*).

## Relative frequency and relative abundance index

The correlation structure of the set of environmental factors (salt lick, wildlife sanctuary guard station, stream, village, road, elevation, and slope) using a matrix of Kendall's rank correlation coefficients, $\tau$ (*Kendall, 1938*; *Halvorsen et al., 2016*) presented by *Chaiyarat et al. (2019)* were used to test the autocorrelation among each environmental factor. Relative frequency (RF) was used to estimate the distribution of the wild Asian elephants, and the relative abundance index (RAI) was used to estimate the abundance of the wild Asian elephants.

## Population survey

Photographs were used to identify and record the location, date, and time of wild elephant sightings. A score of 0 indicated bad image quality and the photograph was discarded. A score of 5–10 indicated that the picture quality was sufficient to identify individual herd characteristics. Photographs' ratings were based on their quality, clarity, and the position of the elephants in the frame (*Varma, Pittet & Jamadagni, 2006*; *Varma, Baskaran & Sukumar, 2012*). A rating of five or above would allow us to determine whether elephants were individual adult males, females, sub-adult males, females, juveniles, or calves (*Arivazhagan & Sukumar, 2008*). We recorded elephants' clear morphological distinguishing features and basic body measurements to help identify individuals (*Goswami et al., 2012*; *Vidya, Prasad & Ghosh, 2014*) (Supplementary S1). Photos were used to identify individual elephants and unique herds. We conducted a census of individual adult males (AM) and adult females (AF), sub-adult males (SM), sub-adult females (SF), juveniles (JU), and calves (CA) for each herd. Herd density was calculated using crude density.
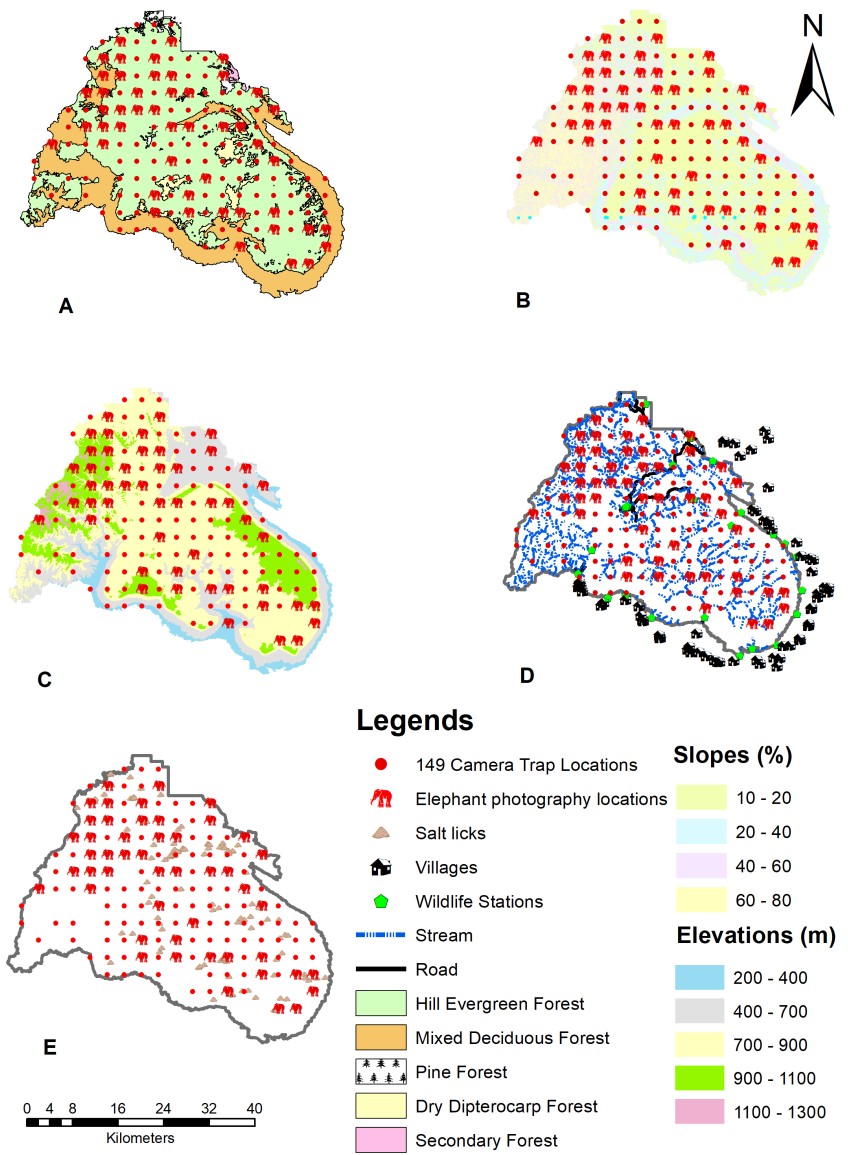

**Figure 2  Environmental factors used to create the distribution model of wild Asian elephants with median grids in the Phu Khieo Wildlife Sanctuary, Thailand.** (A) Forest types. (B) Slope. (C) Elevation. (D) Human activity (villages, roads, wildlife stations, and streams). (E) Salt licks.

## Herd distribution and distribution model

We used camera traps (*Aebischer, Robertson & Kenward, 1993*) to obtain statistically independent wild Asian elephant distribution data. Photographs taken by our traps were used to identify each herd. The location of each herd was used to predict distribution patterns and population habitat with MaxEnt. We only used MaxEnt with species records when individuals were either present, pseudo–absent, or truly absent at any given point on the landscape within a given time frame (*Pearce & Boyce, 2006*).

We used the jackknife procedure and percentage factor contributions to estimate the relative influence of different predictive factors in MaxEnt. Model performance was evaluated using the area under the receiver-operating characteristic (ROC) curve (AUC) (*Fielding & Bell, 1997*) as previously reported in *Chaiyarat et al. (2019)*.

## Population home range

Wild Asian elephants are a herd animal; therefore, we grouped their home ranges for our study. In *Chaiyarat, Youngpoy & Prempree (2015)* and *Chaiyarat et al. (2019)*, we used the kernel density-estimate (KDE) bounds on the innermost 95% of the 49 presence data points to estimate habitat areas (*Seaman et al., 1999*). The model derived from this equation was used to create a habitat use map in ArcGIS 9.3 (Environmental Systems Research Institute; *ESRI, 2007*).

## Statistical analysis

The RF was calculated for all camera trap locations as previously described in *Chaiyarat, Youngpoy & Prempree (2015)*:

RF = No. camera locations that captured photographs $\times 100$/total camera locations

The RAI calculated for all camera trap locations as:

RAI = No. of detections for wild Asian elephants $\times 100$/total number of camera trap nights

Wild Asian elephant detection was considered to be independent if the time between consecutive photographs of the same individual was more than 0.5 h apart. This definition follows (*O'Brien, Kinnaird & Wibisono, 2003*).

The crude density (D) was calculated as:

D = Total number of wild Asian elephants/total number of Phu Khieo Wildlife Sanctuary

The habitat suitability area ranged from 0 (completely unsuitable habitat) to 1 (optimal habitat). Suitable habitats were classified into three categories:

Most suitable = most optimal habitat for wild Asian elephants >0.66 to 1

Moderate suitable = moderate optimal habitat for wild Asian elephants >0.33 to 0.66

Lowest suitable = low optimal habitat for wild Asian elephants >0 to 0.33

In *Chaiyarat et al. (2019)* we used MaxEnt to calculate the AUC value in a slightly different manner (*Phillips, 2017*) an AUC value of 0.5 indicated that the model did not perform better than a random model, whereas a value of 1 indicated perfect discrimination (*Swets, 1988*).

A one-way ANOVA was used to compare the population structure in different areas. The correlation coefficient was used to analyze the relationship between population structure and water sources, natural licks, elevation, slope, and forest types. We used SPSS as previously described in *Chaiyarat, Youngpoy & Prempree (2015)*. Environmental factors affecting population structure were considered to be significant at $p < 0.05$. This work was conducted under an appropriate animal ethics approval (COA. No. MU-IACUC 2016/17) with permission from the Department of National Parks, Wildlife, and Plant Conservation (NRCT No. 0402/3908).

**Table 1 Kendall's rank correlation coefficients between pairs of environmental factors in the Phu Khieo Wildlife Sanctuary.**

| Salt lick | | | | | | |
|---|---|---|---|---|---|---|
| 0.070 | Station | | | | | |
| −0.035 | 0.192** | Stream | | | | |
| −0.110155* | 0.469** | −0.193** | Village | | | |
| 1.000 | 0.070 | −0.035 | −0.110* | Road | | |
| 0.089 | 0.412** | −0.211** | 0.669** | 0.089 | Elevation | |
| 0.229** | −0.093 | −0.039 | −0.155* | 0.229** | −0.025 | Slope |

**Notes.**
The $\tau$ value is always between −1 and 1.
$\tau = 1$, Perfect (very strong) positive correlation.; $\tau = -1$, Perfect (very strong) negative correlation.; $\tau = 0$, zero (no) correlation.
** correlation is significant at the 0.001 level (2-tailed).
* correlation is significant at the 0.05 level (2-tailed). villages, roads, and streams were combined as "Human Activity" in the distribution maps.

## RESULTS

### Autocorrelation test

The correlation structure for seven environmental factors (station and stream, station and village, station and elevation, village and elevation, salt lick and road, and road and slope) had a very strong positive correlation. In contrast, there was a negative correlation between two factors (stream and village; stream and elevation; $t = 0.192$) that was very statistically significant ($p < 0.001$) (Table 1). These environmental factors helped determine the distribution models.

### Relative frequency and relative abundance index

The RF was 32.9% for all camera trap stations and the RAI of wild Asian elephants in the PKWS was 62.6 captures per 100 trap nights (Table 2).

### Population survey

A total of 114 wild Asian elephants were identified from 4,319 photographs taken at 49 camera trap stations in the PKWS (Table 2). The crude density of wild Asian elephants in the sanctuary was 0.07 individuals per km$^2$. The population consisted of 30 adult males, 43 adult females, 14 sub-adult males, nine sub-adult females, 11 juveniles, and seven calves, with the population ratio was 0.7:0.1:0.3:0.2:0.3:0.2. The ratio of reproductive ability between adult females (including sub-adult females), juveniles, and calves was 1.0:0.2:0.1 ($F = 1.072$, $df = 5$, $p = 0.382$) (Table 3).

### Herd distribution

Wild Asian elephants were separated into seven herds based on individual classifications (Fig. 3). Herds 1–5, roamed in the northwestern part of the sanctuary, while the other two preferred the eastern (herd 6), or western regions (herd 7).

### Distribution model

We used MaxEnt to calculate the habitat model of the wild Asian elephants as previously described in *Chaiyarat et al. (2019)*. Our results revealed that all 15 models generated
**Table 2** Relative frequency (RF), relative abundance index (RAI), and the environmental factors affecting elephants in camera trap stations in the Phu Khieo Wildlife Sanctuary between 2016 and 2019.

| Environmental factor | Present | | Trap-night (Nights) | Encounter rate | RAI |
|---|---|---|---|---|---|
| | Number | % | | | |
| Total camera trap station | 149 | 100 | 6,896 | 4,319 | 62.6 |
| Relative frequency (RF) | 49 | 32.9 | | | |
| Forest | | | | | |
|     Dry evergreen forest | 41 | 83.7 | | | |
|     Mixed deciduous forest | 7 | 14.3 | | | |
|     Pine forest | 0 | 0 | | | |
|     Dry dipterocarp forest | 1 | 2 | | | |
|     Secondary forest | 0 | 0 | | | |
|     Total | 49 | 100 | | | |
| Elevation (m) | | | | | |
|     200–400 | 2 | 4.1 | | | |
|     400–700 | 5 | 10.2 | | | |
|     700–900 | 29 | 59.2 | | | |
|     900–1,100 | 7 | 14.3 | | | |
|     1,100–1,300 | 6 | 12.2 | | | |
|     Total | 49 | 100 | | | |
| Slope (%) | | | | | |
|     0–20 | 27 | 55.1 | | | |
|     20–40 | 10 | 20.4 | | | |
|     40–60 | 8 | 16.3 | | | |
|     60–80 | 4 | 8.2 | | | |
|     Total | 49 | 100 | | | |

**Table 3** Population structure and sex ratio of wild elephants in the Phu Khieo Wildlife Sanctuary between 2016 and 2019.

| Herd | Elephant (Individuals) | Individuals (Sex ratio) | | | | | | A | Reproductive ratio | | | B |
|---|---|---|---|---|---|---|---|---|---|---|---|---|
| | | AM | AF | SM | SF | JU | CL | | AF | JU | CA | |
| 1 | 12 | 2(1) | 6(3) | 0 | 2(0.3) | 2(0.3) | 0 | 2.67 | 1 | 0.3 | 0 | 2.67 |
| 2 | 22 | 3(1) | 10(3.3) | 5(0.5) | 2(0.2) | 2(0.2) | 0 | 2.55 | 1 | 0.2 | 0 | 2.55 |
| 3 | 25 | 6(1) | 10(1.7) | 3(0.3) | 2(0.2) | 2(0.2) | 2(0.2) | 2.6 | 1 | 0.2 | 0.2 | 2.6 |
| 4 | 13 | 2(1) | 5(2.5) | 2(0.4) | 1(0.2) | 3(0.6) | 0 | N/A | 1 | 0.6 | 0 | 2.75 |
| 5 | 12 | 4(1) | 6(1.5) | 0 | 1(0.2) | 0 | 1(0.2) | 1.6 | 1 | 0 | 0.2 | N/A |
| 6 | 14 | 3(1) | 4(1.3) | 4(1) | 1(0.3) | 0 | 2(0.5) | 2.54 | 1 | 0 | 0.5 | 2.54 |
| 7 | 8 | 2(1) | 2(1) | 0 | 0 | 2(1) | 2(1) | N/A | 1 | 1 | 1 | N/A |
| Total | 114 | 30(1) | 43(1.4) | 14(0.5) | 9(0.3) | 11(0.4) | 7(0.2) | N/A | 1 | 0.3 | 0.2 | N/A |

**Notes.**
F, 1.072; df, 5; $P$-value, 0.382; A, Duncan test for sex ratio; B, Duncan test for reproductive ratio; N/A, not analyzed.

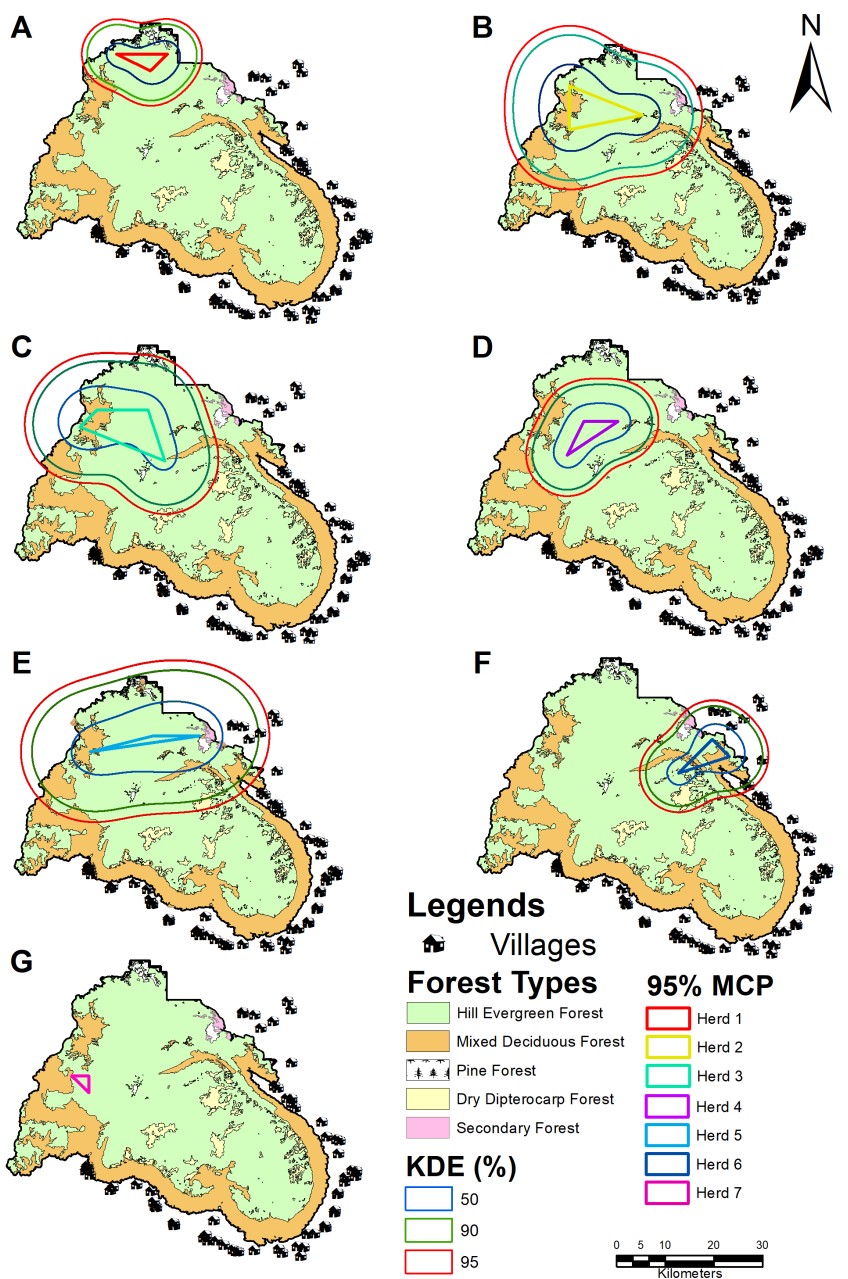

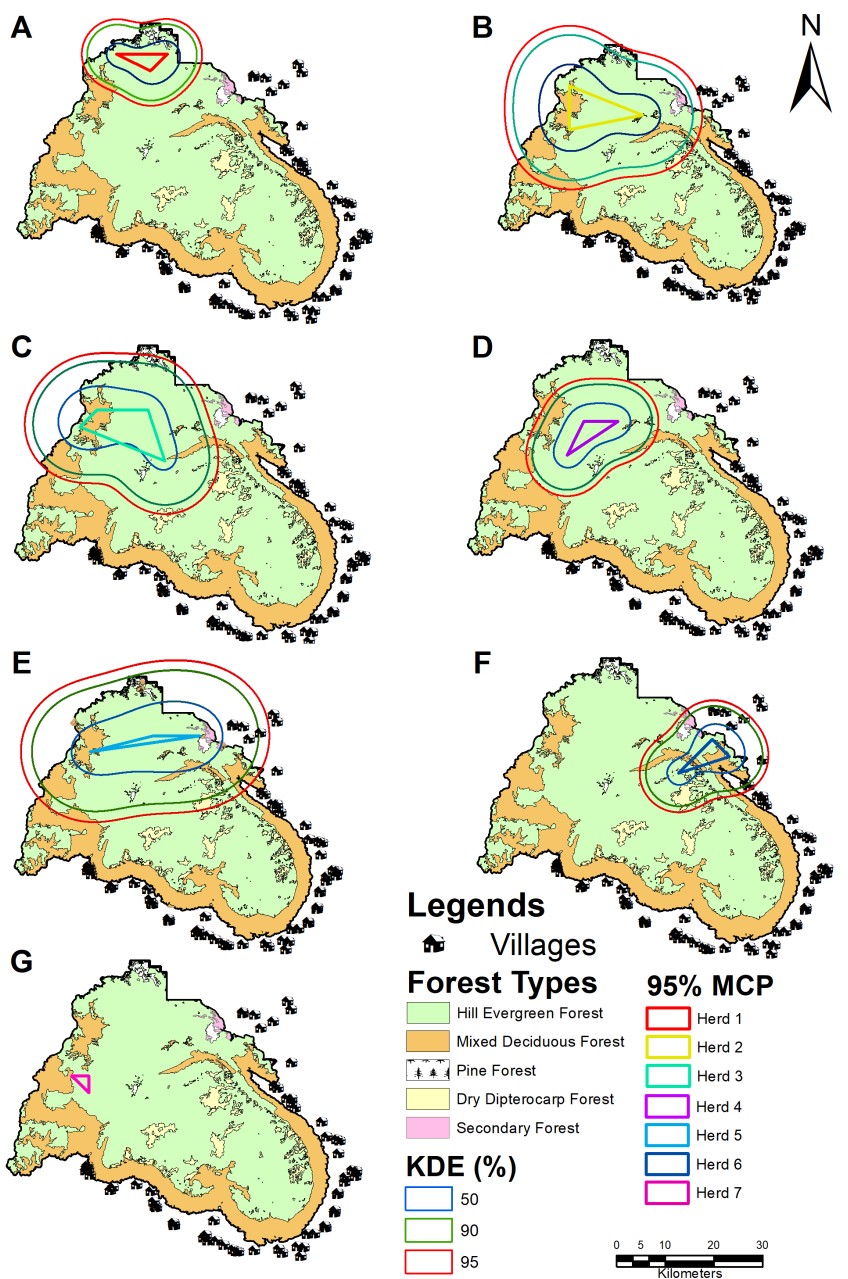

**Figure 3** **Population habitat use of each wild Asian elephant herd in the PKWS between 2016 and 2019 using MCP and KDE.** Herd 1 (A), herd 2 (B), herd 3 (C), herd 4 (D), herd 5 (E), herd 6 (F), and herd 7 (G).

training or testing models when compared with a random model where the average AUC was 0.61 ± 0.13. The test AUC values were lower than the training AUC values (Figs. 4A, 4B). The average training AUC values were 0.689, while the test AUC values ranged from 0.393 to 0.819. The model was run 15 times (Table 4). The contribution of the environmental factors and the results of the jackknife test analysis are presented in

**Table 4 Estimated relative contributions of environmental variables, overall prevalence of training AUC, test gain, and test AUC of the MaxEnt model.**

| Environmental variable | Percentage of contribution (%) | Permutation importance (%) | |
|---|---|---|---|
| Saltlick | 14.9 | 36.4 | |
| Elevation | 52 | 23.5 | |
| Human Activity | 9.8 | 20.2 | |
| Forest | 23.2 | 19.2 | |
| Slope | 0.1 | 0.6 | |
| Model | Training AUC | Test Gain | Test AUC |
| 0 | 0.69 | −0.14 | 0.494 |
| 1 | 0.731 | −1.211 | 0.393 |
| 2 | 0.704 | −0.277 | 0.41 |
| 3 | 0.67 | 0.227 | 0.731 |
| 4 | 0.675 | 0.414 | 0.819 |
| 5 | 0.681 | 0.131 | 0.667 |
| 6 | 0.697 | 0.047 | 0.59 |
| 7 | 0.686 | 0.007 | 0.566 |
| 8 | 0.69 | 0.111 | 0.629 |
| 9 | 0.694 | 0.048 | 0.59 |
| 10 | 0.678 | 0.302 | 0.814 |
| 11 | 0.695 | −0.033 | 0.519 |
| 12 | 0.677 | 0.233 | 0.76 |
| 13 | 0.692 | −0.046 | 0.543 |
| 14 | 0.679 | 0.077 | 0.619 |
| Average | 0.689 | −0.007 | 0.61 |

Fig. 4C. Analysis of the environmental factors independently indicated that distance from salt licks, elevations, land covers, and forest types were more important than slopes. The distance from salt licks was the most important predictor (36.4%) of habitat suitability. The second and third most important factors were elevation (23.5%) and human activities (20.2%), respectively. The contribution of environmental factors (Table 4) and response curves (Fig. 5) showed that the main environmental factors affecting habitat suitability were elevation (52%), forest types (23.2%), and distance from the salt licks (14.9%). Using the MaxEnt habitat model, we determined that species' suitable area was 1,288.9 km$^2$; 276.9 km$^2$ was found to be unsuitable (Table 5). Highly and moderately suitable areas were 672.3 and 616.6 km$^2$, respectively. Most elephant herds were situated between 700-900 m above average sea level (ASL) and found in dry evergreen forests. However, most solitary males and a herd of males were found 400-700 m above the PKWS. The MaxEnt habitat model was similar to the minimum convex polygon (MCP) and 95% KDE. The model showed that wild Asian elephants used a wide range of habitats (Table 5 and Fig. 6). The total area of 95% of the MCP was 1,098 km$^2$. The whole area inside the PKWS using 95% of the KDE was 1,554.9 km$^2$.

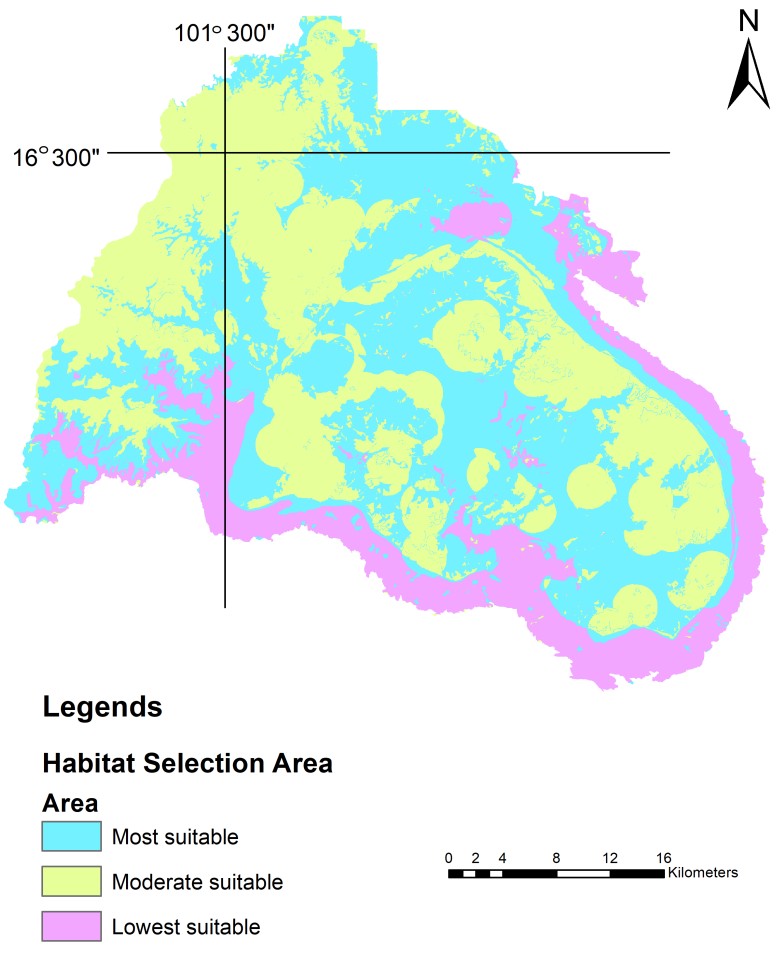

**Legends**

**Habitat Selection Area**

**Area**

- Most suitable
- Moderate suitable
- Lowest suitable

0  2  4  8  12  16 Kilometers

**Figure 4** **Distribution model of wild Asian elephants with median grids in the Phu Khieo Wildlife Sanctuary, Thailand.**

## Population home range

The population of herd 3 was the largest of the seven female herds at 72 km$^2$ (Table 5 and Fig. 3D). The second largest was herd 2 at 67.5 km$^2$ (Fig. 3C). The smallest was herd 7 at only 4.5 km$^2$ (Fig. 3H).

## DISCUSSION

The Asian elephant population in this sanctuary currently has 114 elephants; in our previous study (*Chaiyarat, Youngpoy & Prempree, 2015*) there were at least 181 elephants. The number of elephants was obtained using camera trap analysis in the Salakphra Wildlife Sanctuary, Thailand. In this study, the RAI was higher than that of the Salakphra Wildlife Sanctuary (*Chaiyarat, Youngpoy & Prempree, 2015*). The elephant density found in these studies (0.07 individuals per km$^2$) was the lowest when compared to the Huai Kha Khaeng Wildlife Sanctuary (0.7 individuals per km$^2$) (*Sukmasuang, 2003*), the Khao Ang Rue Nai Wildlife Sanctuary (0.1 individuals per km$^2$) (*Wanghongsa et al., 2006*),
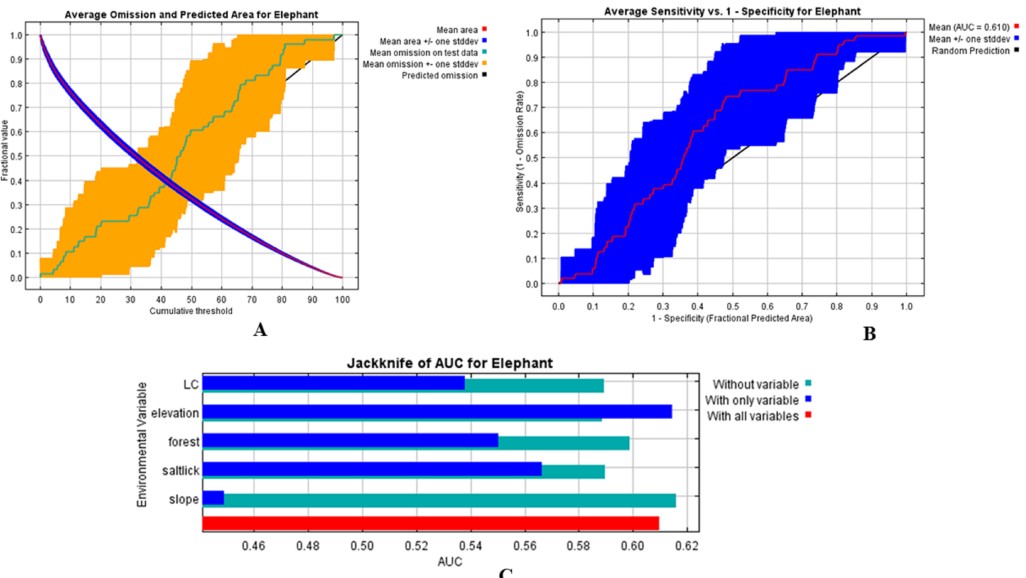

**Figure 5** **Test omission rate and predicted area curves as a function of the cumulative threshold.** Average over the replicate runs (A), curve of the receiver operating characteristic (ROC) plot with evenly spaced thresholds and area under the ROC (AUC) for our habitat suitability model using MaxEnt (B), the average test AUC for the replicate runs is $0.61 \pm 0.13$ (mean $\pm$ SD) and results of the jackknife test for contributions of the variables of the wild Asian elephant habitat model in Phu Khieo Wildlife Sanctuary, Thailand (C). Distance from each land used and land cover (LC) is given in m, elevation is given in m, forest is distance from forest (m), saltlick is distance from salt lick (m), and slope is the percentage of slope (%).

and the Salakphra Wildlife Sanctuary in Thailand (0.21 individuals per km$^2$) (*Chaiyarat, Youngpoy & Prempree, 2015*). It was also lower than the Bardia National Park in Nepal (0.2 individuals per km$^2$) (*Flagstad et al., 2012*). According to *Sukumar & Santipiallai (1993)*'s carrying capacity model, wild Asian elephant density may reach 0.2 to 0.3 individual per km$^2$ or 312 to 468 individuals in the PKWS. This may be the reason that elephants remain inside the sanctuary when compared to other protected areas (*Chaiyarat, Youngpoy & Prempree, 2015*): their population is at 36.5%, the lowest carrying capacity range described by *Wanghongsa et al. (2007)*. During our study, there was no immigration or emigration.

The population structure in the area was comprised of mostly adults, as previously described in *Chaiyarat, Youngpoy & Prempree (2015)*. Obtaining this information was only possible due to camera traps identifying the individuals and classifying their age and sex (*Varma, Pittet & Jamadagni, 2006*). The population also consisted of more adult females than adult males and there were high reproduction rates. Overall, however, the reproductive ratio of wild Asian elephants in the PKWS was relatively low compared to the findings of *Katugaha, De Silva & Santiapillai (1999)* in Ruhuna National Park, Sri Lanka, *Choudhury (1999)* and *Ramesh et al. (2012)* in India, and *Chaiyarat, Youngpoy & Prempree (2015)* in the Salakphra Wildlife Sanctuary, Thailand. In our study larger herds tended to have a higher reproductive ratio when compared to smaller ones, such as herd 7, and may be why the total population in our study was the lowest when compared with others. The

**Table 5  Suitable area (MaxEnt) and home range of wild Asian elephant herds with Minimum Convex Polygon (95% MCP) and Kernel Density Estimate (KDE) in the Phu Khieo wildlife Sanctuary between 2016 and 2019.**

| Population habitat use | Herd No. (km²) | | | | | | | All Herd (km²) | |
|---|---|---|---|---|---|---|---|---|---|
| | 1 | 2 | 3 | 4 | 5 | 6 | 7 | Total area | Inside PKWS |
| Suitable area (MaxEnt) | 13.5 | 67.5 | 72 | 18 | 13.5 | 18 | 4.5 | 1,288.9 | N/A |
| Most suitable | 2.2 | 10.4 | 7.3 | 2 | 9.9 | 9.8 | 4.5 | 672.3 | N/A |
| Moderate suitable | 11.3 | 57.1 | 64.7 | 16 | 3.6 | 2.5 | 0 | 616.6 | N/A |
| Lowest suitable | 0 | 0 | 0 | 0 | 0 | 5.8 | 0 | 276.9 | N/A |
| Forest type | | | | | | | | | |
| Hill evergreen forest | | | | | | | | | N/A |
| Mixed deciduous forest | | | | | | | | | N/A |
| MCP (95%) | 13.5 | 67.5 | 72 | 18 | 13.5 | 18 | 4.5 | 1,098 | N/A |
| Forest type | | | | | | | | | N/A |
| Hill evergreen forest | | | | | | | | | N/A |
| Mixed deciduous forest | | | | | | | | | N/A |
| KDE | | | | | | | | | |
| 95% | 58.9 | 118.2 | 100.1 | 69.9 | 127.1 | 65.9 | N/A | 3,131.9 | 1,554.9 |
| 90% | 53.0 | 106.5 | 90.6 | 62.2 | 114.2 | 59.3 | N/A | 2,522.7 | 1,535.2 |
| 50% | 33.7 | 68.8 | 59.1 | 38.9 | 70.9 | 38.6 | N/A | 8,70.3 | 826.5 |
| Forest type with 95% KED | | | | | | | | | |
| Hill evergreen forest | | | | | | | | | N/A |
| Mixed deciduous forest | | | | | | | | | N/A |

**Notes.**
Habitat suitability area ranged from 0 to 1. Most suitable, most optimal habitat for wild Asian elephants ≥ 0.66 to 1; Moderate suitable, moderately optimal habitat for wild Asian elephants ≥ 0.33 to 0.6; Lowest suitable, suboptimal habitat for wild Asian elephants ≥ 0 to 0.33; N/A, not analyzed.

variability in population dynamics likely reflects differences in environmental conditions and carrying capacities between sites as previously described in *Ramesh et al. (2012)* and *Chaiyarat, Youngpoy & Prempree (2015)*. These conditions include elevation, land covers, and salt licks in the PKWS. We found that the elephant population is increasing in PKWS as previously described in *Chaiyarat, Youngpoy & Prempree (2015)* in the Salakphra Wildlife Sanctuary.

In Indonesia, the Sumatran elephants preferred lower elevation (<200 m) and slopes between 0 to 20% (*Wilson et al., 2021*). Our study found that wild Asian elephants preferred higher elevations, while another suggested that wild Asian elephants generally avoid feeding or walking in upland areas to save energy (*Wall, Douglas-Hamilton & Vollrath, 2006*). Elephants in the PKWS are found at high elevations as the sanctuary is located on the plateau and flat plains on the top of the mountain, with deep slopes along the boarder (*Faculty of Forestry, 2010*). Conversely, solitary male elephants appeared at higher elevations in dry evergreen forests to avoid conflict with the dominant males in the lowland areas (*Chaiyarat, Youngpoy & Prempree, 2015*). This finding is similar to that of *Steinmetz et al. (2008)*, who found elephants in the hilly evergreen forests above 1,000 m in the Thung Yai Naresuan Wildlife Sanctuary, Thailand. *Wanghongsa et al. (2006)* and *Joshi (2009)* documented elephants in areas up to 1,300 m ASL. Wild Asian elephants in the

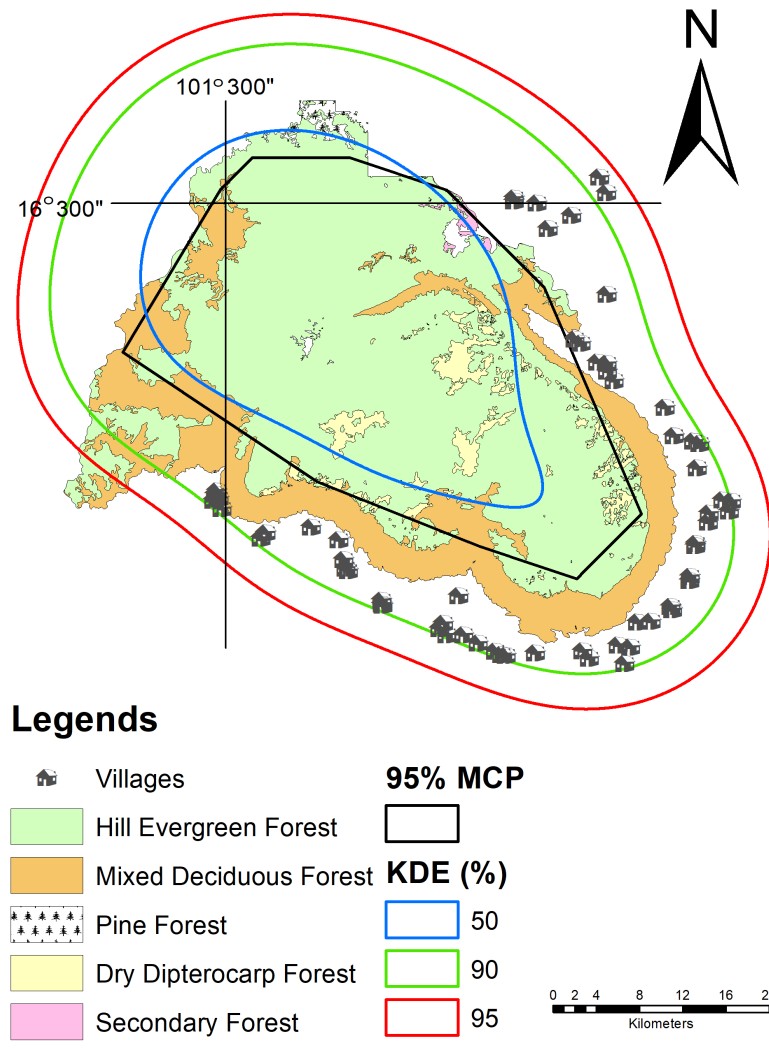

**Legends**

| | | | |
|---|---|---|---|
| 🏠 | Villages | **95% MCP** | |
| 🟩 | Hill Evergreen Forest | ☐ | |
| 🟧 | Mixed Deciduous Forest | **KDE (%)** | |
| ✶ | Pine Forest | ☐ | 50 |
| 🟨 | Dry Dipterocarp Forest | ☐ | 90 |
| 🟪 | Secondary Forest | ☐ | 95 |

0 2 4 8 12 16 20
Kilometers

**Figure 6** **Wild Asian elephant populations'. habitat use in the Phu Khieo Wildlife Sanctuary between 2016 and 2019. Habitat use was determined with the MCP and KDE.**

PKWS mainly used shallow slopes (0–20%), which is consistent with our previous research (*Chaiyarat, Youngpoy & Prempree, 2015*). We also found that elephants also used flat plains, which is consistent with the results of other studies (*Alfred et al., 2012*; *Chaiyarat, Youngpoy & Prempree, 2015*). Waterholes are not a prominent environmental factor in the PKWS (*Alfred et al., 2012*; *Chaiyarat, Youngpoy & Prempree, 2015*) as water can be found in most areas. A large section of the sanctuary was covered by dry evergreen forests which proved to be the most suitable habitat for wild Asian elephants. This is in contrast to the Salakpra Wildlife Sanctuary, where the most suitable habitat for wild Asian elephants are mixed deciduous forests since bamboo, the elephant's favored food, is dominant in this area (*Gray & Phan, 2011*; *Chaiyarat, Youngpoy & Prempree, 2015*). Food-plant productivity is positively related to utilization by elephants *Rood, Ganie & Nijman, 2010*) and the PKWS

contains small bamboo areas. This is a primary factor affecting their movements (*Lin et al., 2011*) and population (*Chaiyarat, Youngpoy & Prempree, 2015*).

Previously we found that wild Asian elephants were photographed more often at salt licks, which were used for nutrient supplementation (*Chaiyarat, Youngpoy & Prempree, 2015*; *Mills & Milewski, 2007*) and the alleviation of gastrointestinal disorders (such as acidosis, diarrhea, and endoparasites) from plant compounds (*Krishnamani & Mahaney, 2000*). Most salt licks were located in proximity to streams or waterholes. The principal factor determining salt lick use was the annual rainfall cycle as elephant movements are strongly controlled by water availability, especially during the dry season (*De Beer & Van Aarde, 2008*).

Our research indicates that the factors relevant to elephant populations are salt licks, elevations, land covers, and forest types. In order to effectively manage wild elephant populations, the following actions must be undertaken: maintain effective salt licks, monitor minerals in the salt licks as they will be beneficial for elephants as well as other wildlife, increase potential food sources such as grassland areas, remove invasive exotic plants and weeds, and reestablish food-plant species in disturbed areas.

*Chaiyarat et al. (2019)* found that the habitat model created using MaxEnt performed well compared with a random model (where the AUC was 0.5). The test AUC values were still lower compared to the training AUC values (*Giovanelli et al., 2010*). Thus, these two models are suitable for studying elephants' habitat suitability. The contribution of environmental factors and results of the MaxEnt jackknife test analysis revealed that the distance from salt licks contained more useful information by itself than the other factors. The next most important factors were elevation and human activity in the PKWS. Using the MaxEnt habitat model, we determined that PKWS elephants' highly and moderately suitable areas were as large as those found in the Salakphra Wildlife Sanctuary (*Chaiyarat, Youngpoy & Prempree, 2015*). Wild Asian elephants in the PKWS selected dry evergreen forests in high elevations, which was different from the habitat used by the population in the Salakphra Wildlife Sanctuary (*Chaiyarat, Youngpoy & Prempree, 2015*). However, this population did not enter agricultural areas (Table 2).

Both data sets calculated using the MCP and KDE (95% KDE; the total area inside PKWS = 1,554.9 km$^2$) are larger than data taken at the Salakphra Wildlife Sanctuary (*Youngpoy, 2012*). Due to the small population (114 individuals), better food quality may have been available in higher quantities than in the larger herds. The suitable-habitat area in the PKWS was also smaller than the one in the Salakphra Wildlife Sanctuary (*Chaiyarat, Youngpoy & Prempree, 2015*). In this study, the MCP was similar to the suitable habitat area (MaxEnt) covering the entire PKWS. In *Chaiyarat et al. (2019)* we found that this trend indicated that an increase in this sanctuary's population may cause animals to enter agricultural areas or other protected area since both areas are suitable to support the population.

In the future, conservation and management should focus on monitoring the population trends, food quality, food quantity, and the physical condition of this population to ensure the long-term conservation of this species. Regular monitoring and surveys are required to build up a comprehensive database on the population trends, improve public awareness

and law enforcement, and effectively manage the habitat. These changes may help reduce the human-elephant conflict in the area.

## CONCLUSIONS

Our study suggests that the wild Asian elephant population in the PKWS was lower than in other areas in Thailand and elsewhere. The resources in the sanctuary are suitable for seven herds. Wild Asian elephant populations in this sanctuary are increasing. In PKWS, wild Asian elephants are distributed according to elevation, the presence of dry evergreen forests, distance from salt licks, and human activity. However, to ensure the long-term conservation of wild Asian elephants and other Asian elephant populations effective management strategies must be used to improve habitat suitability.

## ACKNOWLEDGEMENTS

We would like to thank the director and staff of the Phu Khieo Wildlife Sanctuary, Department of National Parks, Wildlife and Plant Conservation for helping us collect the data.

### Funding
Funding support was received from Mahidol University, Thailand. The funders had no role in study design, data collection and analysis, decision to publish, or preparation of the manuscript.

### Grant Disclosures
The following grant information was disclosed by the authors:
Mahidol University, Thailand.

### Competing Interests
The authors declare there are no competing interests.

### Author Contributions
- Nyi Nyi Phyo Htet and Rattanawat Chaiyarat conceived and designed the experiments, performed the experiments, analyzed the data, prepared figures and/or tables, authored or reviewed drafts of the paper, and approved the final draft.
- Nikorn Thongthip and Panat Anuracpreeda conceived and designed the experiments, authored or reviewed drafts of the paper, and approved the final draft.
- Namphung Youngpoy performed the experiments, prepared figures and/or tables, and approved the final draft.
- Phonlugsamee Chompoopong performed the experiments, authored or reviewed drafts of the paper, and approved the final draft.

## Animal Ethics

The following information was supplied relating to ethical approvals (i.e., approving body and any reference numbers):

Mahidol University-Institute Animal Care and Use Committee (MU-IACUC) provided full approval for this research (COA. No. MU-IACUC 2016/17).

## Field Study Permissions

The following information was supplied relating to field study approvals (i.e., approving body and any reference numbers):

The Department of National Parks, Wildlife and Plant Conservation and National Research Council of Thailand approved this research (NRCT No. 0402/3908).

## Data Availability

Data and raw measurements are available in the Supplemental Files.

## Supplemental Information

Supplemental information for this article can be found online at http://dx.doi.org/10.7717/peerj.11896#supplemental-information.

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
