# Peer review of "Population and distribution of wild Asian elephants (Elephas maximus) in Phu Khieo Wildlife Sanctuary, Thailand"

_PeerJ, doi:10.7717/peerj.11896_

## Round 0.1 · original submission · Major Revisions

Writing in a foreign language is difficult. Might I recommend the program Grammarly? I use it routinely and, I believe, one way or another you might be able to get free access to it. It provides very detailed comments — which means a lot of work — but I also know it can be hard or expensive to find someone to check the writing.

Reviewer 1 ·

Basic reporting

No comment. I included all matters in my report.

Experimental design

No comment. I included all matters in my report.

Validity of the findings

No comment. I included all matters in my report.

Additional comments

Reviewer report
Population and distribution of wild Asian elephants (Elephas maximus) in Phu Khieo Wildlife Sanctuary, Thailand
General comments
I commend the authors in establishing such a rich dataset on elephant locations in a single protected area in Thailand, in just over a 4-year period. I read the manuscript with interest, as I have never been to Thailand, not have I ever seen wild Asian elephants, in their natural environment. So, my review of the manuscript must be taken by the authors as me forming part of an international audience interested in the conservation and protection of wild Asian elephants, in Thailand. I, however, have four broad concerns, which needs to be dealt with first by the authors, in order for them to translate this rich dataset to published information.
First, the Introduction needs to be re-written to answer four basic questions: What is the problem? Why is it important? What have I/we done to address the problem? What do we expect from our study – what is the most salient and significant outcome (as a take-home message for the reader of the paper? Some of these questions has been addressed by the authors i.e., the issues of habitat loss, and landscape transformation (which then were partly addressed in the manuscript), but addressing poaching, and then admit this not being a problem (and not forming a focus point in the paper) therefore needs to be removed from the introduction. The third question – what have I/we done to address the problem need the most attention, because towards the end of the Results section, the manuscript started to fall apart, and I lost track of the flow of the paper (see the third general comment).
Second, English is not my first (or even close to my home) language, and I suspect the same for the authors of this manuscript. I would therefore strongly suggest, as I often having to do, that the authors make use of a professional and/or colleague proficient in English to assist with the grammatical and editorial issues (we as reviewers and Editors can’t do it for the authors), to improve the manuscripts’ readability by removing value-laden jargon currently scattered across the manuscript.
Third, and along with the second point, the authors may want to consider re-structuring the paper. Reading through the Results section of the paper, especially towards the end, I lost the flow of the result presented there, as the description of the results jumped from one set of results to the next, introducing new variables being tested with no mention where these ‘new’ set of variables came from (saltlick, station, stream, village, and road, Dry Dipterocarp forests, although later for the land cover description in the building the MaxEnt models), and leaving out other environmental variables listed earlier in the manuscript (i.e., savanna), and then only to mention (in line 174) threat factors as an explanatory variable(s). Nowhere in the manuscript could I find these threat factors.
I strongly suspect Figure 2 is missing some of the cartograms (and as the information here links with results presented in Table 5) made the whole manuscript fall to pieces towards the end. This is crucial information on the distribution of elephants in the protected area, and needs attention before further effort can be spent on the review-process.
I am, of course, willing to give it another round of review, once these issues has been resolved, as a proper re-written, edited, carefully design manuscript, with the attention to detail it deserves.
Fourth, I don’t see the point in including the sex-specific age-structures of the seven elephant herds in this paper, as the paper is mainly concerned with the distribution of elephants, and the habitat variables associated with such a distribution. Only including the very basic demographic (overtly detailed) structure detracts from the main message of the paper, and serves no purpose.

Minor specific comments
Abstract
From 38,834 photos, 149 locations, 6,896 trap nights yielded the exact same number of 6,896 photos with elephants – is this correct? Remarkable if it is the case.
MaxEnt gives habitat preference, within 49 of the 149 trap locations (in the main text it listed 7, sometimes 5 environmental factors, sometimes other not listed anywhere – see general comments)
Line 45: Commonly referred (insert to) as an umbrella…..
Line 46: remove the word clearings, and change forest to forests
Line 47: reference change (IUCN 2020), surely the IUCN is not the authority to list here
Lines 49 to 54: need References implicating habitat loss, conversion of forests to agriculture, loss of migratory routes, and poaching (both commercial and subsistence)
Line 55: Are these the latest figures available for wild Asian elephants in Asia (see also line 67)?
Lines 58 to 71: is best illustrated with an informative map showing the position of Thailand with its neighbouring countries, the position of the listed protected areas, illustrating the sizes of the protected areas (and complexes), and the distances between them, and then specifically for the study site in question. Figure 1, in its current format does not provide for this, and updating Figure 1, will help in having to remove the unnecessary text in the Introduction. Figure 1 itself is very unclear, the heading and legends not informative for international readers, and the map show that most of the elephant locations (although I am unsure what the elephant symbols means) are situated in the moderate suitable (yellow) area? Or is this an effect of the camera trap locations, and how where these locations selected?
Line 64: it is back to the poaching issue – which apparently is not a threat in Thailand (needs Reference), or removed, as this does not form part of the study (unless, it is one the threat factors, which hasn’t been described anywhere)
Lines 65-66: degradation and fragmentation the biggest threat (needs Reference) – leading to an increase conflict (needs Reference)
Line 66: change ‘A rough estimate …’ to ‘An estimate …’
Line 67: The number of elephants in Thailand 17 years ago not relevant to this study? Are these the latest estimated figure for elephants across Asia, and Thailand?
Line 68: how do we know that the estimate is imprecise? Is that an opinion, or actually published, which then needs a Reference
Lines 68 to 71: again best illustrated in a map (as suggested before)
Lines 68 to 70: these lines are repeated in the study area description, and need to be removed from the introduction
Line 69: what is a ‘best wildlife spot’? For tourism? Or for the protection of biodiversity/Asian elephant? If it is the latter – then the authors need to consider a proper Reference.
Lines 72 to 83: this paragraph reads like an opinion piece, Describing camera techniques being trusted? With ease of use? And identification of species/individuals being unobtrusive, shoulder heights for age, used elsewhere should really form part of the Methods, not the Introduction
Line 88: Remove the sentence on stating the ‘best wildlife spots’
Line 89: the names should be spelled consistently, either Latitude and Longitude, or latitude and longitude
Lines 91 to 93: The study site map (Figure 1) should also illustrate all the detail (i.e., names, locations, areas, (8) connections, etc.,) as described here
Lines 94 to 97: elevation, slope, land cover, and vegetation types are the most salient features (individually, or combined) for elephants which is again, is best illustrated visually with a separate map to the one in Figure 1. Then, on this map the 149 camera trap locations can be indicated. For instance, only reporting on the average and highest elevations, hides information on the (%) low-lying areas, and the associated habitat features for these elephants (as the Results also imply between 200m to 300m elevation band widths). Some 95% of the study area consists of two types of forests, and more details on these are needed. If these forests do not differ substantially, but do dominate the landscape, should then not be considered as an explanatory variable for habitat suitability. If they do differ substantially, then forest type becomes relevant (along with the other landscape features selected). Are the elephant locations the same as trapping station locations?
Lines 97-100: the statement can be removed as I suspect that this is not relevant to the study (but can be included as part of the newly designed study site figure – see comment above)
Lines 102 to 104: Change the entire sentence in its current format to: We used HCO SG565 flash camera-traps (HCO Outdoor Products, Norcross, Georgia, USA) to obtain photographs of individual elephants.
Line 106: remove … about 15 m or even to …
Line 106: remove entirely – we can never say we have captured an entire herd in one photograph, as individuals may have walked by outside the photographic area. I assume these were digital images made of elephants – how frequently were these downloaded from each trapping station?
Lines 107 to 109: move these sentences before where you describe attaching the cameras to trees (and then, also, indicate on the study site map, the locations of the 3 x 3 km2 grids. In addition, how many cameras were used per grid, and how far apart are these cameras placed, and then how many of these 3 x 3 km2 grids as they move to a new location every 3 months?
The results of the set of correlation structure of environmental factors (are reported in Table 1) are new and as of yet, not described (or even better, illustrated). For instance, if the authors could illustrate and describe salt licks (sometimes mentioned natural slat licks, other time only slat licks, what are these and how are they distributed across the study site), station (I presume camera-trap station), show the streams (water being such an important factor for any elephant) and of course with elevation, village (proximity or are these even inside the protected area, size, number of people if there are more than one village in question), illustrate the road network, elevation (already covered), and slop (slope? also mentioned earlier). This is a rather important step in the analyses routine, and your international readers (like me not being familiar with Thailand, or its elephants, and protected areas) will require much more and better illustrated information.
(Table 1) heading and legend reads the same
Lines 111 to 113: Instead of just referring to the work of others, rather provide the full description of the variables included here (see the point above), and then for your own understanding, in just a few words what the purpose of this exercise was. It will then become clear why more information is needed here
Lines 113 to 115: the calculation of relative frequency to estimate elephant distribution has me confused – so, does no photographs illustrate no elephant presence, and relatively high, or low frequency indicate elephant presence? Does relative frequency, the number of times an elephant (herd) walked by a camera, give an indication of elephant distribution? What if it is the same herd (false high frequency), or different herds walking past single and multiple times (false high frequencies), same herds but in different composition over the three years? In principle, therefore, we need to know the degree of observer bias in separating (identifying/distinguishing) among the animals – as you did report in the abstract the exact number of elephants, and their sex-specific and life stage specific numbers (and ratios). We would like to review this (as a suggestion, it can be written up and included as Supplemental Material)
Lines 116 to 121: The relative abundance index, also is depended on the details requested for the relative distribution calculations. You mentioned O’Brien et al. (2003) as a reference of independence of animal detections of the same species were longer than 30 minutes apart but, surely, it should be of the same individual here, as camera-happy (as opposed to camera-shy individuals) will inflate estimates – and may eventually give an incorrect population estimate (and the related population characteristics that you listed in your abstract). If all elephants were treated the same, we cannot expect to see absolute values – and only relative indices
Line 127: change claves to calves
Line 123 to 133: In addition of this description, an illustration of a ‘typical’ (with photographs, and illustrations of the morphological distinguishable and body dimensions) in identifying individual elephants (maybe suggested as part of Supplemental Materials). How did you decide when it was an elephant bull, or elephant cow, or when it was a sub-adult bull? What is the difference (maybe in relative size on photos) the difference between a juvenile and a calf, and a juvenile and a sub-adult (bull or cow)?
Lines 135 to 136: How frequently should these elephants be seen together to make up one herd? And what is the typical composition of a Wild Asian elephant herd, compared to what you have found at your study?
Lines 138 to 139: delete the entire first sentence
Lines 142 to 157: move to earlier in the methods section
In general – the methods and the analyses procedures were described together, which confused me, as I had to waste time to search for the analyses and methods description relevant for each of the result sections.
Line 168: Am I understanding this correctly, calculating a home range based on 49 data points. Is this correct?
Lines 172 to 176: For the Statistical Analysis, this is where the question (what have we done in the Introduction becomes relevant) for I cannot figure out what exactly do the authors want from their work? What is population structure mentioned in line 173? And for the first time the threat factors are included in statistical analyses – what are these threat factors, how were they measured for elephants, and scaled for inclusion in the ANOVA?
Line 182: Given the outcomes of the autocorrelation test – how was the models corrected after excluding some factors, and which ones remained, or were they just all kept irrespective of whether there was a strong or weak negative and positive correlation?
Line 193: How was crude density calculated?
Table 3: I would suggest, rather to give the absolute values, to illustrate the population structure and sex ratios, because we have to calculate backwards the composition]. For the reproductive ratio the values are fine. Present the total at the bottom of the table, i.e., first herd 1-7, then total values for the whole population. Also, describe the Duncan test in the analyses description. Change N/A = not analysis to N/A = not analysed
Lines 199 to 207: Unnecessary information, duplicating everything in Table 3, can be deleted
Table 2 comes into play. And this is where restructuring the paper may help the reader.
Lines 209 to 211: appears to be misplaced
Lines 212 to 215: the result description, and the incomplete cartograms in figure 2 does not tell me much, apart from a relatively high degree of overlap, and therefore, the total value results in Table 5 does not make sense to me. I cannot figure out where the total values (I think it is in km2) comes from – which is one of the main results reported in the abstract, and therefore must be important. Are all the values of each herd used independently, and then added?
Table 4: change slop to slope: What are models 5 to 14? And the average – the ‘lack-box calculations here needs much information to be evaluated.
Table 5: Note:-=Not predicted should maybe read Note: N/A not analysed?
First mention of MCP. And then for KDE, are the total just added up, but what about the overlap among the elephant herds? I guess, with Figure 2 incomplete, and Table 5 linked to Figure 2, must first be addressed before carrying on.
In table 5: we need more information on what is ‘most suitable’, ‘moderate suitable’, ‘lowest suitable’ for wild Asian elephants.
In Figure 2: each diagram needs a number, I expected 8 cartograms, and only see 7
In Figure 2: some of the KDE 95% green lines shave different thickness – what does the size differences mean?
In Figure 2: Please check if Kernal should not be Kernel
In Figure 2: I can’t find the MCP lines?
In Figure 2: A space should follow after F.
In Figure 2: The figure title stops abruptly – and then appears to be repeated as the description below it – this problem appears in some of the other Table and Figure descriptions – and should be addressed with more attention to detail.
Line 236 to 243: now I completely lost track
Discussion (I did not read the discussion)

Reviewer 2 ·

Basic reporting

The text is understandable, but the English is sometimes difficult to follow.

The context is clear, the issue also, the literature is relevant. It is very focused on South-East Asian elephant though.

The structure conforms with PeerJ, that said figures are not very neat, though acceptable and readable.

Raw data is supplied, and appendix table open easily.

Recommendation: A native English proof-reading may improve the fluidity of the manuscript. A better lay-out of figures would be needed to conform to PeerJ standards but does not preclude understanding.

Experimental design

The study is in the scope of PeerJ, the interest is moderate in terms of knowledge gaps, but it has value for conservation of the species. Some details are missing on the area considered as a sample unit around the camera traps, and this should be addressed to allow for a strict replication of the study.

The use of MaxEnt modelling is rudimentary, and some conclusions therefore shallow (see next point). My advice here would be to rerun the analyses, and use a model selection approach to select a parsimonious model.

Validity of the findings

The analyses are run correctly but the model is very weak, hence the conclusions are overstated. Further, the conclusions may have overlooked a simple ouput of the MaxEnt model: elevation is the only variable that performs better than the model with all variables, suggesting that a simple model with elevation may be enough. This relates to possible covariation with elevation of other key variables.

As stated in point 2, I would rerun the model and use consider model selection, and certainly focus on a limited set of variables. The jack-knife procedure is used for this partially, and strongly suggest that elevation alone is enough.

Additional comments

Although the paper presents interesting data on a small population of Asian elephant, and certainly contributes to a better understanding of its status in the area, there are several issues regarding the description of the method and the use of the MaxEnt modelling that prevent the conclusions to be very powerful, and also question the relevance of some of the variables considered, as discussed in points 2 and 3. See detailed comments enumerated below, hoping it will help addressing the issues.

L69-70 I think the structure of the sentence calls for correction. I suggest add [is]  as follow : Phu Khieo Wildlife Sanctuary, one of the best wildlife spots in Thailand, [is] located in Chaiyaphum province in Western Isaan Complex in the northern part of Thailand.

L144-146 “...with a set of environmental factors that likely model a species’ environmental requirements were used from set of occurrence localities, influence the suitability of the environment for the species...”. What is the distance around the camera trap that was used as a spatial sampling unit? Was this unit matching the pixel size used for slope or elevation? This needs to be clarified.
L148: ‘Furthermore,..’ this word is not necessary, as the authors presnt ways to extract variables enumerated previously.
L154-155: Is this sentence meaning that only 49 camera sites captured elephants and the other 100 did not, hence being considered as pseudo-absence (or real absence) ? ‘A model was built by using 49 wild Asian elephants’ 155 coordinates (6,896 photographs) from the total 149 camera trap stations (38,834 photographs).

L172-174: The sentences describe very rudimentary analyses, only using one-way anova and correlations. The data seem to allow a bit more elaborated statistics.
L175: “Differences between population structure and environmental factors were considered significant at p < 0.05.” I believe should read “Effects of environmental factors on population structure were considered significant at p < 0.05.”

Results

L201-205: The following information is redundant with Table 3. “…population structure among AM:AF:SM:SF:JU:CA of herd No. 1 was 1:3.0:0.0:0.3:0.3:0.0; herd No.2 was 1:3.3:0.5:0.2:0.2:0.00, herd No. 3 was 1:1.7:0.3:0.2:0.2:0.2; herd No. 4 was 1:2.5:0.4:0.2:0.6:0.0; herd No. 5 was 1:1.5:0.0:0.2: 0.0:0.2; herd No. 6 was 1:1.3:1.0:0.3:0.0:0.5 and herd No. 7 was 1:1.0:0.0:0.0:1.0:1.0. The sex ratio between AM:AF was highest in herd No. 2 (1:3.3) and lowest in 205 herd No. 7 (1:1.0).”

L218-219: I think the authors are over emphasizing the significance of their models, as the 0.5 value is included in the interval around the average: “...15 generated training or testing models (for all records modelled) had a high level of performance compared with a random model where the average AUC was 0.61±0.13.” Further, Fig 3B shows that the random line is mostly included in the confidence interval of the sensitivity.

Table 1: I wonder why slope is not at all correlated with elevation, it is often the case, except on high altitude plateau. The correlation can be weak but rarely have I seen it not correlated at all. That echoes another point raised by the examination of the first suppl. file, in which lots of values for elevation are 0 (actually 172 ‘0’ for only 18 non-zero values). I think it might not have any relevance then, which is actually suggested by the MaxEnt analysis.

Table 5 and results on this section: I’m not sure the comparison are relevant as information are not really of the same type. The result do give an idea of how much range seems favorable to elephant, and the difference between herds. Vulnerabilities of herds with home ranges using less favorable habitat may be more interesting to show.

---

## Round 0.2 · Minor Revisions

As you can see, the reviewer has a long list of suggestions (s)he wishes you to make.

My concerns are that your writing still needs substantial improvement. Now, I have many colleagues and students whose first language is not English. I understand how difficult it is to write in a second language and I suggested you use a top-rated program such as Grammarly to help you. PeerJ does not copy-edit your text as part of the standard service. Likely what you submit next time will be accepted and published as-is. I urge you to find better ways to improve what you have written.

Reviewer 1 ·

Basic reporting

Basic reporting (Mainly editorial)
Abstract
Conclusions
Line 40: change ‘, the habitat is able to provide a suitable habitat area for elephants, …’ to ‘, this Sanctuary is able to provide suitable habitat for elephants, …’
Line 42: change ‘area’ to ‘region’
Line 42: change ‘reduce the conflicts between’ to ‘reduce conflict between’ and remove ‘along with the broader areas.
Introduction
Line 54: change ‘threaten their populations as well’ to ‘threaten some populations'
Line 55-57: the sentence here reads confusing: it reads as if the elephants increased from 41,410 to 52,345 in a single year, and then to decline in numbers by 2020. Consider rewriting the sentence to make the statement clear for the reader

The tables and figures have many spelling mistakes. I.e., Table 1 where Slop should be Slope
Table 2: would it be possible to get clarity on when the authors talk about a grid, a camera trap station, etc?
In general, the figures are a great improvement from the first round of review.
Figure 2: (D) Human activity and streams map, I assume the black line(s) are people presence, activity, or villages. Further clarity is needed here, as sometimes people (presence) are referred to as activity (legend of figure 2), or villages in map 2(D) and Table 1, in the result section village and elevation, are seemingly combined into one of seven environmental factors, the presence of people (or their activities not even included in Table 4; and then referred to as village location in figure 6). And yet it is highlighted as one of the main issues in the Introduction (lines 50-55) where it refers to potential human-elephant conflict. All of the people aspects needs better clarification, or if not an issue, removed.

Experimental design

Experimental design (again, mainly editorial)
Materials and methods
Study area
Line 81-82: slopes range from 10% to 80% - what do these percentages mean?
Camera trap Survey
Lines 87-91: please consider splitting these very long (and repetitive sections) into two, possibly even three shorter sentences.
Lines 105-117: The whole paragraph here is out of place, and not even needed as it has nothing to do with the study design.
Lines 118-127: This entire paragraph confuses me: in a 3 x 3 km2 grid, and two cameras per grid (camera station I presume) gives 18 cameras per grid, not 15 as stated. The sentence on ‘The spatial sampling unit…..’ is misplaced, and according to the authors the batteries have been downloaded (probably meaning the batteries have been replaced) and the SD cards downloaded.
Population survey
Line 139: the scale needs to be quantified; i.e., 0 means bad quality photos and discarded, and 5-10 as sufficient from which to identify individual (herd) characteristics
Lines 150-153: This section can be incorporated into the next (Distribution Model) section.
Lines 155-156: remove the entire sentence
Lines 154-172: this is the crux part of the methods section in the paper, but it reads so confusing, and with irrelevant information that has nothing to do with the survey design, that I cannot figure out if the study was done correctly.
Statistical analysis
The authors keep on mentioning ‘species’ but surely you mean ‘individuals’, as this work is on a single species, but numerous individuals?
Line 186: what does PKWS mean?
Lines 194-197: does this section relate to the previous description in lines 188-193? If so, then this needs to be clarified and rewritten
Validity of the findings
It is unclear if both the positively and negatively correlating factors were included to estimate the distribution model(s)? The description of the correlating factors is very confusing.
Distribution Model
Line 236: ….that all 15 generated … 15 what? Models? It is not clear
Lines 243: what does more useful mean?
Line 253-254: the last sentence reads like a conclusion, and not as a result per se
Lines 255-258: here the authors write about a trend, of the MaxEnt habitat model and the MCP and KDE? What trend do you refer to here? As the home range results are only indicated in the next section

Validity of the findings

No comment.

Additional comments

General comments
The Introduction still lacks a clear problem statement, apart from stating some generalities that are applicable in any part of the world with elephants, and people nearby. In general, having read through the manuscript it would appear that the authors have not considered a professional editorial (grammar) input, as was previously suggested. Past, present and future tenses are all over the show! The whole manuscript is written in a passive voice and detracts from the readability of the paper. Some of the sentences are seven lines long, and by the end of each sentence I forgot what the beginning was – it also causes confusion by repeating statement(s) in a single sentence. This needs immediate attention. Having read through the methods and analyses section, I am still not sure what the questions are that the study wants to answer, and the flow of information disjointed and difficult to follow.

Conclusion
The conclusion section only reflects, and repeat the main findings (which also, was never made clear in the result section) with limited deductive reasoning of what this study contributes to our broader understanding of wild Asian elephants, their management, and conservation. Statements like in lines 367-370 are dangerous, in that here the authors reflect on controlling the reproductive ratio of the population. Not enough information is provided by the authors to make such a suggestion.

---

## Round 0.3 · Minor Revisions

I still cannot accept your paper in its current form. PeerJ does NOT copy edit manuscripts. Yours still has problems with its English. The first sentence of the abstract is missing a verb, for instance. It should read

The populations of wild Asian elephants (Elephas maximus) **are** increasing in many protected areas in this decade after a period of worldwide decline.

It's essential that you get your manuscript checked by someone who has excellent English. I'm sorry to be so critical, but we can't publish this until you. Other than the English, I have no other concerns.

---

## Round 0.4 · accepted · Accept

Thank you for making the changes!